# Differences in Demographics of Vaccinees, Access to, and Satisfaction with SARS-CoV-2 Vaccination Procedures between German General Practices and Mass Vaccination Centers

**DOI:** 10.3390/vaccines10111823

**Published:** 2022-10-28

**Authors:** Anne Jentzsch, Anne-Kathrin Geier, Markus Bleckwenn, Anne Schrimpf

**Affiliations:** Department of General Practice, Faculty of Medicine, Leipzig University, Philipp-Rosenthal-Str. 55, 04103 Leipzig, Germany

**Keywords:** COVID-19, vaccines, immunization programs, preventive health services, primary health care, vaccination coverage, mass vaccination, vaccination hesitancy, health services accessibility

## Abstract

In the European Union, SARS-CoV-2 vaccines became available in December 2020. The vaccination campaign in Germany was initially implemented through mass vaccination centers and later joined by general practitioners (GPs) in spring 2021. This study compared population characteristics, perceived access barriers, and satisfaction with the vaccination procedure between vaccination centers and GP practices. A paper-based survey was distributed (07/2021–10/2021) among newly vaccinated individuals in ten GP practices (*n* = 364) and two vaccine centers (*n* = 474). Participants in vaccine centers were younger compared to participants in GP practices. GP preference was higher in older participants and those with pre-existing illnesses. Wait time at vaccination site was longer in GP practices, whereas travel distance to site was longer for participants in vaccine centers. However, satisfaction with patient education and recommendation of site were more likely with increasing comprehensibility of the vaccination procedure and physicians’ information as well as perceived sufficiency of patient education duration, factors that can be easily modified by all vaccination sites. Our results demonstrate that both types of vaccination sites complement each other in terms of accessibility and target population and that satisfaction with the vaccination procedure can be promoted at all sites by an easy-to-understand process.

## 1. Introduction

Since the outbreak of the COVID-19 pandemic in 2020, exceptional investments have been made worldwide to counter the novel disease. An essential role to combat COVID-19 has been ascribed to the development of effective vaccines and their quick roll-out [1]. The urgency of the situation provided vaccine manufacturers with enormous funding, less bureaucracy, and, thus, the possibility to run multiple clinical trials in parallel [2,3]. As early as December 2020, the first SARS-CoV-2 vaccine was authorized in the European Union, followed by two more vaccines in January 2021 [4].

Aiming at an efficient roll-out of the initially tight vaccine margins, most European countries, including Germany, prioritized individuals with a high risk of contracting COVID-19 or having a more severe course of the disease [5,6]. In addition, logistical challenges arose with the distribution and storage of mRNA vaccines [7,8], which is why mass vaccination centers were implemented to manage the national strategy. In Germany, 474 vaccine centers were established between December 2020 and May 2021 [9], with the largest of these capable of administering up to 10,000 doses per day [10]. In April 2021, general practitioners (GPs) across Germany were allowed to join the vaccination campaign in their private practices [11], which significantly accelerated the vaccination roll-out [12]. Both vaccination sites (vaccine centers and GP practices) were not restricted in vaccine types in Germany. In the vaccine centers, all authorized vaccine types were vaccinated, which were allocated centrally depending on availability. GPs could order vaccines according to their demands, but only in limited quantities. The vaccine doses were provided by the federal states and funded by the German government [13]. Officially, vaccinees could not choose the vaccine type. However, the mRNA vaccines from BioNTech or Moderna were used more frequently due to growing safety concerns about AstraZeneca’s vaccine, resulting in its approval only for individuals over 60 years of age since the end of March 2021 [14].

The hybrid solution of vaccinations in both GP practices and vaccine centers appeared to be the key for rapid nationwide coverage. It has been discussed that GPs and mass vaccination centers might complement each other and reach different groups of the population [12,15]. Vaccine centers, on the one hand, have been centrally located to immunize large numbers of people [16], potentially leading to regional disadvantage for people living in rural areas, who often have to travel long distances for their vaccination appointments [17]. In addition, the requirement to register in the online portal to schedule a vaccination appointment might have limited accessibility for disadvantaged groups such as individuals with restricted internet access, non-native speakers, or older people [18]. In contrast, GP practices offered the advantage of being more evenly distributed across the country [15], making their access more convenient, especially for individuals with reduced mobility. Further, the familiar environment and patients’ high levels of trust in GPs might increase the acceptance of vaccination even in vaccination hesitant groups [19], wherefore they might for some groups be the preferred option for vaccine administration [20].

Although differences in access between GP practices and vaccine centers are well known [15,18,19], to our knowledge, no study directly compared population characteristics, perceived barriers to access, and satisfaction with the vaccination procedure between these two types of vaccination sites in Germany. The aim of our study was, hence, to distribute a paper-based survey among newly vaccinated individuals in both GP practices and vaccine centers. Besides sociodemographic characteristics and their influence on preference, we assessed whether access to the vaccination site (e.g., time between scheduling the appointment and vaccination date, travel distance, means of transport, barriers) differed between GP practices and vaccine centers and how these differences would affect satisfaction with the procedure. We also examined if the duration of the educational interview and the reported content and quantity of the information given by the physicians before vaccination differed between sites and how these differences would influence satisfaction with the patient education. Finally, potential factors influencing the willingness to recommend the vaccination site were explored.

## 2. Methods

### 2.1. Sampling and Design

The cross-sectional survey was carried out in the Free State of Saxony, Germany. Based on the overall population in Saxony (4,043,002 in 2021), a percentage of occurrence of 50%, confidence level of 95%, and a 4% margin of error, we calculated a required sample size for surveys of at least 601 completed questionnaires. Data were collected between July 2021 and October 2021. However, most of the questionnaires were collected in July (76.5%) and August (12.2%). Vaccinated individuals older than 18 years of age completed the survey immediately after receiving their first or second SARS-CoV-2 vaccine within their recommended 15 min observation time. Vaccine centers and GP practices were invited on a convenience basis to voluntarily participate in this study and to hand out a self-developed questionnaire to recently vaccinated individuals. The questionnaires were distributed in vaccine centers and GP practices in city and in rural areas. The two vaccine centers received in total *n* = 650 questionnaires: *n* = 350 in Leipzig-Messe (city area) and *n* = 300 in Belgern (rural area, Nordsachsen). Ten GP practices received in total *n* = 450 questionnaires, of which five GP practices were located in the city area of Leipzig (*n* = 195 questionnaires) and five practices were located in a rural area, Leipziger Land/Nordsachsen (*n* = 255 questionnaires). The recruitment process can be found in Figure 1.

### 2.2. Questionnaire

The patient questionnaire was self-developed in the Department of General Practice of the Leipzig University by an interdisciplinary research team (medical scientists and GPs) in a multi-stage revision process. The questionnaire development was also based on an extensive literature search aiming at identifying relevant factors for the satisfaction with vaccination procedures. The questionnaire was adjusted for the respective vaccination site and varied in terms of wording. It is composed of the following topics: (1) socio-demographics (e.g., age, gender, education, occupation, place of residence, family status), (2) medical history (e.g., pre-existing diseases, previous infections with the coronavirus), (3) information on the current SARS-CoV-2 vaccine (e.g., vaccine type, date of first and/or second dose), (4) vaccination access (e.g., wait time for appointment, registration, means of transport, traveling time), (5) vaccination procedure (e.g., wait time in vaccination site, quality and quantity of information given on side effects), and (6) satisfaction with the vaccination site.

Education was assessed by using the CASMIN educational classification [21]. Attitudes and satisfaction were assessed by means of either 5-point or 10-point rating scales. Participants were asked on a 5-point scale if they agree or disagree with a given statement (e.g., “The distance I had to travel to get vaccinated was too far”. 1 = Strongly disagree, 5 = Strongly agree). Item-specific response options were given on 10-point rating scales (e.g., “Were you generally satisfied with the patient education?”, 1 = Not at all satisfied, 10 = Very satisfied).

The questionnaire underwent a think-aloud pre-testing [22] aimed at identifying problems or misunderstandings related to each item and was adjusted afterwards. The provisional questionnaire was filled out by five patients from the targeted group who were instructed to think aloud while answering each item and report every spontaneous thought. All patients were additionally interviewed after the session. This led to two items being added and seven items being revised and simplified. The final version of the questionnaires can be found in Appendix A.

### 2.3. Ethics Statement

The study was carried out in accordance with the Declaration of Helsinki and the study protocol was approved by the research ethics committee of the Leipzig University (reference number 314/21-ek). Participants agreed to participate by voluntarily returning the anonymous questionnaire. No personal data besides age and gender were assessed.

### 2.4. Statistical Analyses

All statistical analyses were carried out using IBM SPSS Statistics 27 (Armonk, NY, USA) with a two-sided α-level of 0.05. For descriptive statistics, missing values in single variables were considered by presenting frequencies as % (*n*/*n*_valid_). Continuous variables were presented as mean ± standard deviation (SD). Group differences in categorial variables were analyzed using chi-square or Fisher’s exact tests, respectively. Estimated effect sizes were reported using Cramer’s V.

Group differences in continuous variables of the patient reports—including age and rating scales—were analyzed using univariate analyses of variance (ANOVAs). Further, the influence of participants’ characteristics (age and pre-existing illnesses) on preference for their vaccination site or a desired vaccination site was analyzed using univariate analyses of covariance (ANCOVAs), with rating scales as dependent variables, age as covariate, and pre-existing illnesses (yes, no) as independent variable. Estimated effect sizes are reported using partial eta squared (η_p_^2^). In case ANOVAs were indicating a significant main effect, least significant differences tests were utilized to determine the origin and direction of the effect.

Further, multiple linear regression analyses using Entry method were conducted. (1) The association of demographic variables, perceived sufficiency and duration of the educational interview, quantity of vaccination information given, and perceived comprehensibility of information (independent variables) with satisfaction with patient education (dependent variable) were calculated. (2) The association of demographic variables, access to site, comprehensibility of vaccination procedure and physicians’ information, perception of patient education duration, and post-vaccination waiting period (independent variables) with the recommendation of vaccination at patients’ vaccination site (dependent variable) were calculated. For both regression models, the assumptions of linearity, residual normal distribution, residual variance homogeneity, residual independency, and no multicollinearity were tested by scatter and *p*-*p* plots, Durbin–Watson statistics, and variance inflation factor (VIF). Cook’s distance was used to detect outliers.

## 3. Results

### 3.1. Sample Characteristics

In total, 1100 questionnaires were distributed in two vaccine centers and ten GP practices (see Figure 1), of which 838 were filled out and were eligible for analyses (response rate of 76.2%). Mean age was 42.5 years and 51.4% of participants were female. Percentages, means, and standard deviations for sample characteristics can be found in Table 1. Besides general characteristics of participants, the table also contains differences in sample characteristics between vaccination sites.

### 3.2. Differences between Sites in Preference, Administering of, and Access to Vaccines

We assessed information on vaccine type, number of doses received, and access to the vaccination site (time between scheduling the appointment and vaccination date, travel distance, means of transport). Percentages and differences between vaccination sites can be found in Table 2.

Additionally, we examined participants’ general satisfaction with the vaccination process. The analyses showed that perceived difficulty getting a vaccination appointment did not differ between sites (*F*(3, 785) = 1.363, *p* = 0.253, η_p_^2^ = 0.005). Participants in vaccine centers perceived the travel distance as further away compared to participants in GP practices (*F*(3, 778) = 24.840, *p* < 0.001, η_p_^2^ = 0.087). Further, participants in rural GP practices perceived the access to the practice as less barrier-free compared to the other sites (*F*(3, 758) = 5.707, *p* < 0.001, η_p_^2^ = 0.022). Participants in city GP practices perceived the vaccination process as slightly more stressful than participants in the rural vaccine center or in rural GP practices (*F*(3, 818) = 3.056, *p* = 0.028, η_p_^2^ = 0.011). Additionally, in the city vaccination center, participants perceived the post-vaccination waiting period as slightly less comfortable compared to the other sites (*F*(3, 814) = 4.792, *p* = 0.003, η_p_^2^ = 0.017). However, all participants would equally recommend the vaccination at their site (*F*(3, 800) = 2.398, *p* = 0.067, η_p_^2^ = 0.009; Figure 2).

Further, we assessed participants’ characteristics (age and pre-existing illnesses) and their influence on preference for their chosen vaccination site as well as their desired vaccination site (see Appendix A) using ANCOVAs. Participants in GP practices rated the importance to be vaccinated by a GP higher with increasing age (*F*(1, 280) = 26.449, *p* < 0.001, η_p_^2^ = 0.086), while pre-existing illnesses were not related to the importance to be vaccinated by a GP (*F*(1, 280) = 1.996, *p* = 0.159, η_p_^2^ = 0.007). In addition, the willingness of participants in GP practices to receive a vaccination in a vaccination center declined with increasing age (*F*(1, 264) = 7.041, *p* = 0.008, η_p_^2^ = 0.026) and with pre-existing illnesses (*F*(1, 264) = 8.953, *p* = 0.003, η_p_^2^ = 0.033, Figure 3A). Importantly, participants in vaccination centers reported with increasing age that they would have preferred to be vaccinated by a GP (*F*(1, 400) = 20.433, *p* < 0.001, η_p_^2^ = 0.049), while pre-existing illnesses were not related to their preference (*F*(1, 400) = 0.438, *p* = 0.508, η_p_^2^ = 0.001, Figure 3B). However, the willingness of participants in vaccination centers to go to a vaccine center again for immunization was not related to age (*F*(1, 401) = 1.600, *p* = 0.207, η_p_^2^ = 0.004) or to pre-existing illnesses (*F*(1, 401) = 0.410, *p* = 0.522, η_p_^2^ = 0.001).

### 3.3. Differences between Sites in Physicians’ Provision of Vaccination Information

In addition, we assessed the content and quantity of information given by the physicians as well as the duration of patient education before vaccination. Percentages, means, and standard deviations for these variables as well as differences between vaccination sites can be found in Table 3.

Further, we examined whether the information was easy to comprehend by the participants as well as the overall satisfaction with the given information. The analyses showed that the comprehensibility of the vaccination procedure did not differ between sites (*F*(3, 796) = 0.251, *p* = 0.860, η_p_^2^ = 0.001). However, comprehensibility of physicians’ information differed between sites (*F*(3, 794) = 5.718, *p* < 0.001, η_p_^2^ = 0.021), indicating that participants in rural GP practices understood the physicians’ information better than participants in city GP practices. In line, participants in rural GP practices perceived the duration of the patient education as more sufficient than participants in the other sites (*F*(3, 800) = 13.238, *p* < 0.001, η_p_^2^ = 0.047, Figure 4). In general, participants in rural GP practices and in the rural vaccine center overall felt more satisfied with the patient education compared to participants in city GP practices and in the city vaccine center (*F*(3, 816) = 8.609, *p* < 0.001, η_p_^2^ = 0.031; Figure 5).

### 3.4. Satisfaction with Patient Education and Recommendation of Vaccination at Patients’ Vaccination Site

A multiple linear regression was calculated to assess relationships between satisfaction with patient education and independent variables such as perceived sufficiency, duration, quantity of vaccination information, perceived comprehensibility of information, and demographic variables. The model explained 37.1% of the variation in satisfaction with patient education (*F*(8, 649) = 47.750, *p* < 0.001; Table 4).

Further, a multiple linear regression was calculated to assess relationships between recommendation of vaccination at patients’ vaccination site and independent variables such as access to site, comprehensibility of vaccination procedure and physicians’ information, perception of patient education duration and post-vaccination waiting period, and demographic variables. The model explained 53.4% of the variation in recommendation of vaccination (*F*(13, 583) = 51.387, *p* < 0.001; Table 5).

For both models, assumption checks for multiple linear regressions were performed before interpretation. Results of these checks can be found in Appendix A.

## 4. Discussion

In our study, vaccinated individuals in German GP practices (*n* = 364) and mass vaccination centers (*n* = 474) were compared in terms of population characteristics, perceived barriers to access, and satisfaction with the vaccination procedure. Participants in vaccine centers were younger and had a higher educational level compared to participants in GP practices. Wait time at vaccination site was slightly longer in GP practices, whereas travel distance to site was longer for participants in vaccine centers. Preference for GP practices was higher in older participants and in participants with pre-existing illnesses. Nevertheless, satisfaction with patient education was not influenced by vaccination site, but by comprehensibility of physicians’ information and perceived sufficiency of patient education duration. Further, although participants in GP practices were more likely to recommend their site, recommendation was strongly influenced by comprehensibility of the vaccination procedure, comprehensibility of physicians’ information, and perceived sufficiency of patient education duration, factors that can be easily modified by all vaccination sites. Our results demonstrate that both types of vaccination sites complement each other in terms of accessibility and target population and that satisfaction with the vaccination procedure can be promoted at all sites by an easy-to-understand process.

### 4.1. Participants’ Socio-Demographic Characteristics and Their Influence on Preference

We examined differences in populations between the two types of sites as well as participants’ preference for one site. We found that participants in vaccination centers were younger, less likely to be retired or to have pre-existing illnesses, and more likely to have a higher level of education compared to participants in GP practices. As we assessed our data in summer 2021 and touristic traveling required immunization in most cases, it is conceivable that younger people increasingly registered at the vaccination centers in preparation for the upcoming holiday. In addition, registration in vaccine centers required a prior online registration, which might have been an obstacle for older people with limited internet access [23,24]. Further, previous studies showed that younger individuals were less likely to have a regular GP [25], which might explain their vaccination site choice.

With respect to differences in education, we found that the proportion of participants with tertiary education was particularly high in the urban vaccine center and, in comparison, much lower in urban GP practices. In contrast, this effect was absent in rural areas, with no differences in educational level between the rural vaccination center and GP practices. This effect might be driven by a generally younger population in urban areas [26], which has been shown to have increasingly higher levels of education [27]. Studies suggested that individuals with a higher educational level were more likely to have a positive attitude towards vaccination [28,29]. Thus, they potentially did not need to be persuaded by a GP to be vaccinated and might have a higher willingness to independently schedule an appointment at the vaccine center. The rural vaccine center, on the other hand, was located in a region in the Free State of Saxony particularly known for their massive undersupply of GPs. We argue that GPs’ reduced availability in this region might have resulted in less individual choice of the desired vaccination site and, thus, increased socio-demographic diversity, which potentially explains the absence of educational differences between GP practices and vaccine center in rural areas.

Importantly, we found that participants with pre-existing illnesses in GP practices reaffirmed their preference for getting vaccinated by their GP more strongly than participants without pre-existing illnesses in GP practices. This effect was absent in participants in vaccination centers. The results might be explained by a generally higher GP consultation rate in patients with chronic conditions and, therefore, a stronger familiarity with their GP and the procedures in the practice [30]. Further, we found in both sites that with increasing age the preference for a SARS-CoV-2 vaccination administered by a GP increased, indicating that older people favored either to be vaccinated in a familiar setting [19] or shorter travel distances [17]. A study from UK found that even in younger individuals GPs were the preferred vaccinators, however, the vaccine center option had no deterrent effects on their willingness to receive the vaccine [20]. Our data suggest that younger individuals who received their vaccine from a GP were more willing to also get vaccinated in a vaccine center compared to older individuals, indicating that in future mass vaccination events younger people could be more easily assigned to mass vaccination centers, whereas older individuals and/or those with pre-existing illnesses should have GP options to increase vaccine uptake.

### 4.2. Access to Vaccination Site

We further compared the access to both types of vaccination sites and individuals’ perception of the access. Our results showed that reported travel time was longer for individuals who received their vaccination in vaccine centers compared to GP practices, which was also reflected in participants ratings: travel distance was more likely perceived to be too long in individuals in vaccine centers. In line, most people used public transport or cars to get to vaccination centers, whereas individuals in GP practices more often walked or rode a bike. Shorter distance to vaccination sites has been proposed to be crucial to increase vaccination uptake [31,32] and should ideally not exceed 30 min [20]. In our study, GP practices were able to offer closer proximity as well as access without car or public transport, which might increase vaccination coverage especially in older individuals, individuals with comorbidities, or with restricted mobility. In contrast, vaccination uptake in mass vaccination centers more likely required the use of a car or public transport, placing people with restricted mobility or lower income at a disadvantage [9,33,34]. In particular, the vaccination center in the rural area in our study was mainly reached by car, potentially driven by poor public transport infrastructure. Our results indicate that vaccination centers, especially in rural areas, might be prone to access inequalities, as has been shown in previous studies [33]. This emphasizes the vital role of GPs as a complement in the national vaccination strategy.

### 4.3. Received Vaccine Type and Dose Intervals

Most respondents received mRNA vaccines. Although all vaccination sites were allowed to administer all vaccine types, depending on availability, AstraZeneca vaccines were administered slightly more frequently in GP practices than in vaccination centers, which was probably related to older vaccinees in GP practices. Since the end of March 2021, AstraZeneca was only approved for individuals over 60 years of age in Germany [14].

Additionally, we assessed the time between the reported administration of the first and second vaccine dose. Recommended intervals between the first and second dose were 3–6 weeks for mRNA vaccines and 12 weeks for AstraZeneca’s vaccine [14]. We found longer intervals for both vaccine types (mRNA and vector vaccine) in GP practices as compared to vaccine centers. Our results indicate that the web-based scheduling of appointments in vaccine centers ensured better compliance with recommended dosing intervals. However, current research indicates that extended intervals offered the same effective immunization against COVID-19 as the initially recommended intervals [35,36]. It can therefore be expected that differing dose intervals in GP practices and vaccination centers have no impact on patients’ immunization.

### 4.4. Physicians’ Provision of Vaccination Information

We analyzed differences between sites in quantity and quality of provided vaccination information. We found that participants in GP practices stated being more often informed about the administered vaccine type, vaccine benefits, as well as vaccine effectiveness compared to participants in vaccine centers. Further, reported duration of patient education was longer in GP practices. Especially in rural GP practices, participants rated the comprehensibility of physicians’ information as higher and the duration of the patient education as more sufficient compared to the other sites. Our results indicate that GPs might be better able—due to their long-term doctor-patient relationships—to estimate the patients’ needs for information. In turn, a more detailed patient communication might increase satisfaction with the provider [37] as well as vaccine uptake [38], especially in individuals whose initial vaccination intention might be uncertain [39]. Rural GPs’ relationships with their patients have been described to be particularly close and more trusting compared to urban GPs’ [40], which might explain higher satisfaction with rural GPs compared to urban GPs [41]. We argue that a potentially closer GP-patient relationship and familiarity in rural areas might both encourage patients to ask for more details about the vaccine as well as support the GP providing the most important information to the individual patient and thus alleviating patients’ uncertainty.

### 4.5. Satisfaction with Patient Education

We also examined the influence of demographics, vaccination site, comprehensibility, content, quantity, as well as duration of information given by the physicians before vaccination on satisfaction with the patient education. We found that overall satisfaction was higher with increasing age and lower levels of education, whereas vaccination site or area had no significant influence. However, satisfaction was strongly influenced by the comprehensibility of physicians’ information and perceived sufficiency of patient education duration. In addition, the quantity of vaccination information given (e.g., vaccine effectiveness, common vaccination reactions, potential complications) was positively related to satisfaction. Importantly, the duration of patient education in minutes was not associated with satisfaction with patient education. In line with our findings, physicians’ communication skills have been shown to increase overall satisfaction in hospitalized patients [42] and might further increase vaccine uptake [43], compliance, and health status [44]. In addition, physicians have been found more likely to have patient-centered interactions with older patients, which might in turn increase the older patients’ satisfaction with the encounter [45]. In sum, our results suggest that satisfaction with the provided vaccine information is less likely determined by vaccination site or familiarity with the provider, but rather by the provision of sufficient and comprehensible information.

### 4.6. Recommendation of Vaccination Site

Lastly, we assessed the likelihood that participants would recommend their site by exploring potentially influencing factors. Our results showed that participants in GP practices were more likely to recommend their site. Recommendation was also associated with perceived sufficiency of patient education duration. The strongest factors for recommending the vaccination site, however, were both comprehensibility of the vaccination procedure and comprehensibility of physicians’ information. Interestingly, demographics, inconveniences such as travel distance or wait time at site, or the perception of the post-vaccination waiting period had no significant influences on recommendation. Although time spent in the waiting room has been found to decrease the likelihood that patients would recommend a practice or care provider [46], other studies did not find this relationship [47,48], indicating that time is not the most influencing factor for recommendation. Indeed, shared decision-making, the physician-patient relationship, as well as easy-to-understand communication were strongly associated with the willingness to recommend the practice or care provider [49,50,51]. These factors might be more likely met by GPs than providers in vaccine centers. However, independent of site, our results suggest that the likelihood of recommendation can be increased by improving the comprehensibility of the procedure and the patient information.

### 4.7. Limitations

Our study has limitations. The data were assessed when the prioritization of specific groups was already suspended, which implies that socio-demographic variables in vaccine centers and GP practices might have differed at the beginning of the immunization campaign. In particular, the survey period coincided with the summer holiday season 2021, potentially leading to more younger people receiving a vaccination to meet travel regulations. Further, participants did not fill in the questionnaire at the same timepoint, instead ranging from July 2021 to October 2021. Temporary differences, such as case incidences of COVID-19, subjective perception of the pandemic’s progression, or holiday seasons, might have influenced participants’ reports over time. In addition, answers regarding travel or wait time and content of the patient information interviews were self-reports and might be imprecise due to subjective perceptions. Further, our questionnaire is not a valid scale as we did not develop and assess several items measuring a construct related to access or satisfaction with vaccination procedures, but rather investigated single item responses.

Lastly, the study was conducted in one federal state in Germany. Differences (e.g., in socio-demographics) between federal states in Germany as well as between European countries limit the generalizability of our findings. Further, the varying access to mass vaccination centers/GPs, availability of vaccines, case incidences, vaccination willingness, and/or potentially still existing prioritization groups in Europe and other parts of the world might impede comparability.

### 4.8. Implications for Practice

Our study offers insights into vaccination preferences of different sociodemographic groups and identifies main factors influencing satisfaction with the vaccination procedure. Our results indicate that, although some groups might prefer a GP or a vaccination center, satisfaction with the patient education as well as the willingness to recommend the site were more strongly influenced by easy-to-understand procedures and information and perceived sufficient duration of doctor-patient conversations. In contrast, travel distance and wait time might not be the most salient factor influencing satisfaction with the vaccination procedure. These results suggest opportunities for future vaccination campaigns but also for other medical procedures: Comprehensibility can be improved in all vaccination sites by using easily understandable language and by implementing linear patient flow and signages to avoid confusion and guide the vaccinees through the process [31,52]. Additionally, providing sufficient time for patient education, offering a broad spectrum of vaccination information, and tailoring the provided vaccination information to individual needs is highly beneficial to increase satisfaction.

However, our results also indicate that access to vaccination centers, especially in rural areas, might put specific population groups at a disadvantage (e.g., those with restricted mobility or lower income). By including GP practices and medical specialists, and later pharmacies [53,54] and mobile vaccination centers (e.g., busses), in the national vaccination campaign, better access to vaccines has been achieved, especially for vulnerable groups, which potentially enhanced vaccine uptake in these groups. For future mass vaccination events, we therefore recommend quickly including both centrally located mass vaccination centers and an appropriate number of decentrally located vaccination sites, such as GP practices, pharmacies, or small, widely distributed vaccine centers in different residential areas.

## 5. Conclusions

We confirm that demographic variables differed between participants receiving their vaccination against SARS-CoV-2 in GP practices and in mass vaccination centers, potentially driven by differences in access requirements, such as travel distance, means of transport, or registration. In addition, GP preference was strongly related to older age, indicating that in future mass vaccination events, allocation to vaccine centers or GP practices could be based on age groups to increase vaccine uptake. However, we found evidence that satisfaction with the patient education and recommendation of the vaccination site can be promoted at all sites by providing sufficient and comprehensible information about the vaccine and the procedure. Our findings provide insights into how future vaccination campaigns or other medical procedures could be designed to achieve higher satisfaction and to optimally meet individual needs in all population groups.

## Figures and Tables

**Figure 1 vaccines-10-01823-f001:**
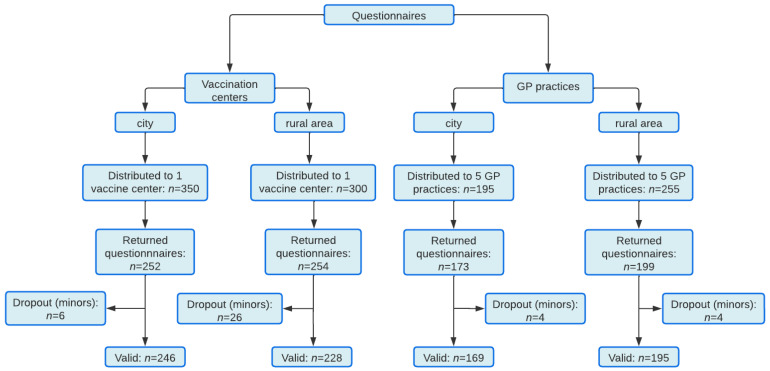
Flowchart of recruitment process.

**Figure 2 vaccines-10-01823-f002:**
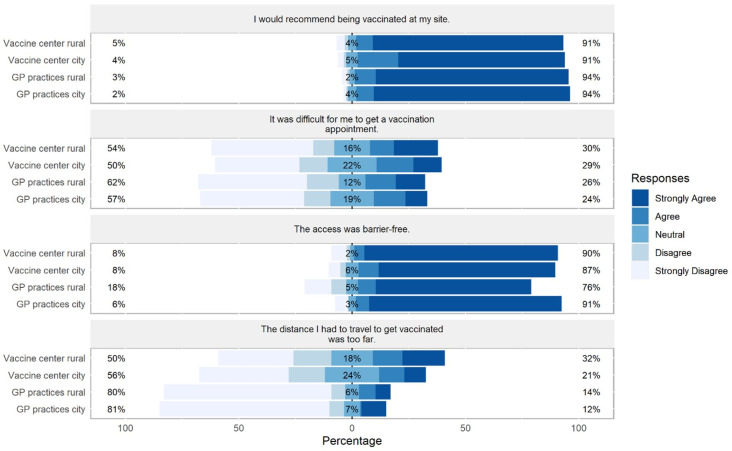
Differences between sites in participants’ responses on preference and access to vaccines. Percentages on the left and right side of the graph represent a summary of Strongly Agree and Agree on the right and Disagree and Strongly Disagree on the left side.

**Figure 3 vaccines-10-01823-f003:**
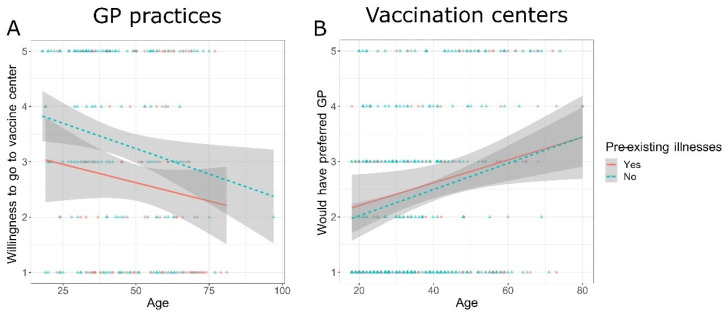
Preference for GPs at both types of sites. Ratings ranged from 1 = Strongly Disagree to 5 = Strongly Agree. (**A**) Participants in GP practices: ANCOVAs revealed that the willingness to receive the vaccine at a vaccination center declined with increasing age of participants (*F*(1, 264) = 7.041, *p* = 0.008, η_p_^2^ = 0.026) and with pre-existing illnesses (*F*(1, 264) = 8.953, *p* = 0.003, η_p_^2^ = 0.033), indicating that pre-existing illnesses influence the willingness to go to a vaccine center independent of age. (**B**) Participants in vaccination centers: The preference to be vaccinated by a GP was higher in older participants (*F*(1, 400) = 20.433, *p* < 0.001, η_p_^2^ = 0.049), independent of pre-existing illnesses (*F*(1, 400) = 0.438, *p* = 0.508, η_p_^2^ = 0.001).

**Figure 4 vaccines-10-01823-f004:**
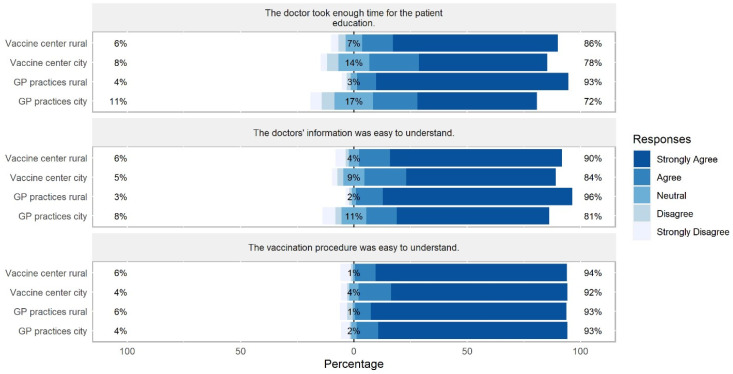
Differences between sites in participants’ perception of comprehensibility and duration sufficiency of the provided vaccination information. Percentages on the left and right side of the graph represent a summary of Strongly Agree and Agree on the right and Disagree and Strongly Disagree on the left side.

**Figure 5 vaccines-10-01823-f005:**
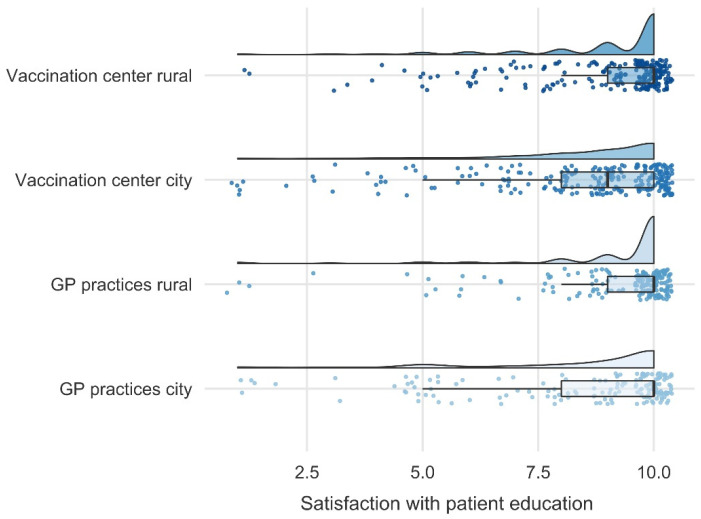
Differences between sites in participants’ overall satisfaction with the patient education ranging from 1 = Not at all satisfied to 10 = Very satisfied. The ANOVA revealed that participants in rural GP practices and in the rural vaccine center felt more satisfied with the patient education compared to participants in city GP practices and in the city vaccine center (*F*(3, 816) = 8.609, *p* < 0.001, η_p_^2^ = 0.031). In general, satisfaction was high in all sites.

**Table 1 vaccines-10-01823-t001:** Sociodemographic sample characteristics and comparison between vaccine sites.

	Total	Vaccine Center City	Vaccine Center Rural	GP Practices City	GP Practices Rural	Comparison between Sites
*n*	838	246	228	169	195	
Age	42.5 ± 16.1	36.2 ± 14.1	40.5 ± 14.4	48.5 ± 17.6	48.1 ± 15.6	*F*(3, 711) = 27.663
						*p* < 0.001
						η_p_^2^ = 0.105
Gender						
female	406 (51.4)	119 (50.0)	107 (50.2)	93 (57.8)	87 (48.9)	χ^2^(6) = 5.574
male	380 (48.1)	117 (49.2)	104 (48.8)	68 (42.2)	91 (51.1)	*p* = 0.411
diverse	4 (0.5)	2 (0.8)	2 (0.9)	0 (0.0)	0 (0.0)	*V* = 0.063
Education						
Primary	10 (1.2)	1 (0.4)	3 (1.3)	2 (1.2)	4 (2.2)	χ^2^(6) = 24.960
Secondary	560 (68.4)	143 (58.4)	159 (70.7)	130 (78.8)	128 (69.6)	*p* < 0.001
Tertiary	249 (30.4)	101 (41.2)	63 (28.0)	33 (20.0)	52 (28.3)	*V* = 0.124
Employment						
Employed	563 (68.1)	176 (71.5)	175 (77.1)	99 (59.3)	113 (60.8)	χ^2^(6) = 54.942
Unemployed	132 (16.0)	45 (18.3)	38 (16.7)	18 (10.8)	31 (16.7)	*p* < 0.001
Retired	131 (15.9)	25 (10.2)	14 (6.2)	50 (29.9)	42 (22.6)	*V* = 0.182
Pre-existing illness(es)	235 (28.8)	44 (18.0)	52 (23.4)	75 (45.7)	64 (34.4)	χ^2^(3) = 42.985
						*p* < 0.001
						*V* = 0.229
Recently infected by COVID-19						χ^2^(3) = 14.211
	29 (3.5)	5 (2.0)	4 (1.8)	4 (2.4)	16 (8.4)	*p* = 0.002
						*V* = 0.147
COVID-19 infection of friends/family	538 (65.1)	167 (68.2)	161 (70.9)	90 (53.9)	120 (63.8)	χ^2^(3) = 13.760
						*p* = 0.003
						*V* = 0.129

Note. Values for age represent mean and standard deviation. Values of other items represent *n* and percentage of valid cases (%).

**Table 2 vaccines-10-01823-t002:** Access to vaccines and comparison between vaccine sites.

	Total	Vaccine Center City	Vaccine Center Rural	GP Practices City	GP Practices Rural	Comparison between Sites
Vaccination no.						χ^2^(3) = 29.187
1	147 (18.0)	51 (21.2)	31 (13.8)	12 (7.4)	53 (27.9)	*p* < 0.001
2	670 (82.0)	190 (78.8)	193 (86.2)	150 (92.6)	137 (72.1)	*V* = 0.189
Vaccine received today						
BioNTech	600 (73.7)	125 (52.1)	189 (83.6)	141 (86.5)	145 (78.4)	χ^2^(9) = 267.673
AstraZeneca	58 (7.1)	3 (1.3)	0	17 (10.4)	38 (20.5)	*p* < 0.001
Moderna	153 (18.8)	112 (46.6)	36 (15.9)	3 (1.8)	2 (1.1)	*V* = 0.326
J&J	3 (0.4)	0	1 (0.4)	2 (1.2)	0	
Vaccine received 1st time (if applicable)						
BioNTech	419 (63.7)	83 (43.9)	150 (80.2)	126 (82.9)	60 (46.2)	χ^2^(6) = 206.724
AstraZeneca	155 (23.6)	52 (27.5)	8 (4.3)	25 (16.4)	70 (53.8)	*p* < 0.001
Moderna	84 (12.8)	54 (28.6)	29 (15.5)	1 (0.7)	0	*V* = 0.381
Same vaccine 1st and 2nd time						χ^2^(3) = 64.341
	529 (82.5)	136 (72.7)	177 (95.7)	132 (91.7)	84 (67.2)	*p* < 0.001
						*V* = 0.314
Days between 1st and 2nd dose	38.3 ± 21.6	36.2 ± 19.0	24.6 ± 9.3	42.4 ± 20.8	58.8 ± 23.2	*F*(3, 610) = 88.658
						*p* < 0.001
						η_p_^2^ = 0.304
Days between 1st and 2nd dose depending on vaccine type						
Both mRNA	27.6 ± 9.1	26.1 ± 6.3	23.2 ± 7.0	33.8 ± 9.5	34.9 ± 10.0	*F*(3, 591) = 9.042
At least one AstraZeneca	70.6 ± 16.1	62.3 ± 15.7	50.0 ± 9.0	79.1 ± 14.7	77.2 ± 11.5	*p* < 0.001
						η_p_^2^ = 0.044
Registration at vaccine center	n.a.			n.a.	n.a.	
Online		189 (78.4)	156 (70.0)			χ^2^(4) = 8.605
Via phone		20 (8.3)	25 (11.2)			*p* = 0.069
By friends or family		12 (5.0)	21 (9.4)			*V* = 0.137
Initiative by center		2 (0.8)	7 (3.1)			
Other		18 (7.5)	14 (6.3)			
Vaccination initiative by	n.a.	n.a.	n.a.			
GP				20 (12.8)	31 (16.8)	χ^2^(3) = 8.734
Myself				105 (67.3)	120 (64.9)	*p* = 0.033
Family/				18 (11.5)	30 (16.2)	*V* = 0.160
friends						
Other				13 (8.3)	4 (2.2)	
Time between scheduling and vaccination date						
<1 week	234 (29.5)	71 (30.3)	71 (32.6)	52 (32.5)	40 (22.1)	χ^2^(12) = 13.471
1–2 weeks	230 (29.0)	69 (29.5)	63 (28.9)	41 (25.6)	57 (31.5)	*p* = 0.337
3–4 weeks	180 (22.7)	56 (23.9)	48 (22.0)	34 (21.3)	42 (23.2)	*V* = 0.075
1–2 months	84 (10.6)	20 (8.5)	17 (7.8)	23 (14.4)	24 (13.3)	
>2 months	65 (8.2)	18 (7.7)	19 (8.7)	10 (6.3)	18 (9.9)	
Wait time at vaccine site						
<10 min	565 (70.4)	200 (83.7)	178 (80.5)	98 (62.4)	89 (47.8)	χ^2^(9) = 85.529
10–30 min	196 (24.4)	36 (15.1)	36 (16.3)	50 (31.8)	74 (39.8)	*p* < 0.001
31–60 min	36 (4.5)	2 (0.8)	6 (2.7)	8 (5.1)	20 (10.8)	*V* = 0.191
>60 min	6 (0.7)	1 (0.4)	1 (0.5)	1 (0.6)	3 (1.6)	
Travel distance						
<5 min	94 (11.4)	3 (1.2)	6 (2.6)	44 (26.8)	41 (21.6)	χ^2^(12) = 344.726
5–10 min	155 (18.9)	21 (8.7)	10 (4.4)	53 (32.3)	71 (37.4)	*p* < 0.001
11–20 min	186 (22.6)	68 (28.2)	43 (18.9)	40 (24.4)	35 (18.4)	*V* = 0.366
21–30 min	142 (17.3)	69 (28.6)	34 (15.0)	19 (11.6)	20 (10.5)	
>30 min	245 (29.8)	80 (33.2)	134 (59.0)	8 (4.9)	23 (12.1)	
Arrival at site						χ^2^(3) = 44.975
Alone	495 (62.1)	141 (60.5)	100 (45.7)	117 (72.7)	137 (74.5)	*p* < 0.001
Accompanied	302 (37.9)	92 (39.5)	119 (54.3)	44 (27.3)	47 (25.5)	*V* = 0.238
Transport						
Walking	95 (11.6)	1 (0.4)	1 (0.5)	53 (32.3)	40 (20.9)	χ^2^(12) = 314.261
Bike	57 (7.0)	20 (8.3)	3 (1.3)	11 (6.7)	23 (12.0)	*p* < 0.001
Car	550 (67.1)	138 (57.6)	214 (95.5)	79 (48.2)	119 (62.3)	*V* = 0.349
Public transport	110 (13.4)	80 (33.3)	1 (0.5)	20 (12.2)	9 (4.7)	
Other	7 (0.9)	1 (0.4)	5 (2.2)	1 (0.6)	0 (0.0)	

Note. Values represent mean, standard deviation, as well as *n* and percentage of valid cases (%). n.a., not assessed.

**Table 3 vaccines-10-01823-t003:** Provision of vaccination information and comparison between vaccine sites.

	Total	Vaccine Center City	Vaccine Center Rural	GP Practices	GP Practices	Comparison between Sites
				City	Rural	
Information given by doctor						
						χ^2^(3) = 8.642
Vaccine type information	503 (62.4)	137 (58.3)	131 (58.2)	104 (65.4)	131 (70.1)	*p* = 0.035
						*V* = 0.104
						χ^2^(3) = 23.866
Vaccine benefits	283 (35.1)	69 (29.4)	60 (26.7)	71 (44.7)	83 (44.4)	*p* < 0.001
						*V* = 0.172
						χ^2^(3) = 28.294
Vaccine effectiveness	423 (52.5)	104 (44.3)	104 (46.2)	88 (55.3)	127 (67.9)	*p* < 0.001
						*V* = 0.187
						χ^2^(3) = 1.072
Behavior before/after vaccination	455 (56.5)	139 (59.1)	126 (56.0)	88 (55.3)	102 (54.5)	*p* = 0.785
						*V* = 0.036
						χ^2^(3) = 7.975
Common vaccination reactions	727 (90.2)	211 (89.8)	206 (91.6)	135 (84.9)	175 (93.6)	*p* = 0.053
						*V* = 0.099
						χ^2^(3) = 3.958
Potential complications	371 (46.0)	97 (41.3)	106 (47.1)	73 (45.9)	95 (50.8)	*p* = 0.273
						*V* = 0.070
						χ^2^(3) = 3.597
Other	13 (1.6)	2 (0.9)	4 (1.8)	5 (3.1)	2 (1.1)	*p* = 0.321
						*V* = 0.067
Quantity of vaccine information given	3.3± 1.8	3.1 ± 1.7	3.2 ±1.6	3.4 ± 2.0	3.7 ± 2.0	*F*(3, 834) = 3.741
						*p* = 0.011
						η_p_^2^ = 0.013
Duration of patient						
education						
<2 min	122 (15.5)	54 (22.9)	31 (13.9)	26 (17.0)	11 (6.3)	χ^2^(9) = 85.069
2–5 min	424 (53.8)	146 (61.9)	133 (59.6)	52 (34.0)	93 (52.8)	*p* < 0.001
6–10 min	206 (26.1)	33 (14.0)	51 (22.9)	58 (37.9)	64 (36.4)	*V* = 0.188
>10 min	36 (4.6)	3 (1.3)	8 (3.6)	17 (11.1)	8 (4.5)	
Possibility to ask questions after vaccination						
Yes	298 (36.6)	74 (30.8)	73 (32.3)	54 (33.5)	97 (51.9)	χ^2^(9) = 33.622
Yes, but not needed	411 (50.5)	135 (56.3)	122 (54.0)	80 (49.7)	74 (39.6)	*p* < 0.001
Not known	58 (7.1)	23 (9.6)	14 (6.2)	13 (8.1)	8 (4.3)	*V* = 0.117
No	47 (5.8)	8 (3.3)	17 (7.5)	14 (8.7)	8 (4.3)	

Note. Values for continuous variables represent mean and standard deviation. Values of categorial variables represent *n* and percentage of valid cases (%).

**Table 4 vaccines-10-01823-t004:** Multiple regression analysis predicting satisfaction with patient education.

Predictor	*B*	*SE B*	β	*R* ^2^
				0.371
Constant	4.214	0.510		
Age	0.015	0.004	0.126 **	
Education	−0.311	0.120	−0.082 *	
Vaccine center/GP practice	−0.118	0.128	−0.031	
Rural/city	−0.244	0.119	−0.065	
Comprehensibility of physicians’ information	0.527	0.104	0.263 **	
Perceived sufficiency of patient education duration	0.496	0.101	0.265 **	
Duration of patient education	0.175	0.090	0.070	
Quantity of vaccine information given	0.076	0.037	0.070 *	

Note. Durbin-Watson = 1.935, * *p* < 0.005, ** *p* < 0.001.

**Table 5 vaccines-10-01823-t005:** Multiple regression analysis predicting recommendation of vaccination at patients’ vaccination site.

Predictor	*B*	*SE B*	β	*R* ^2^
				0.534
Constant	1.221	0.240		
Age	0.000	0.001	0.006	
Education	−0.068	0.044	−0.045	
Vaccine center/GP practice	0.122	0.050	0.079 *	
Rural/city	0.056	0.044	0.037	
Perceived distance to vaccination site	−0.008	0.016	−0.016	
Access to site was barrier-free	0.038	0.023	0.055	
Wait time at vaccine site	−0.010	0.039	−0.008	
Comprehensibility of the vaccination procedure	0.340	0.033	0.368 **	
Perceived sufficiency of patient education duration	0.093	0.039	0.123 *	
Duration of patient education	−0.062	0.034	−0.062	
Comprehensibility of physicians’ information	0.263	0.040	0.327 **	
Quantity of vaccine information given	−0.012	0.014	−0.027	
Perception of the post-vaccination waiting period	0.026	0.014	0.055	

Note. Durbin-Watson = 1.872, * *p* < 0.005, ** *p* < 0.001.

## Data Availability

The data that support the findings of this study are available on request from the corresponding author, A.J.

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
