# Peer review of "Differences in Demographics of Vaccinees, Access to, and Satisfaction with SARS-CoV-2 Vaccination Procedures between German General Practices and Mass Vaccination Centers"

_vaccines, 2022, doi:10.3390/vaccines10111823_

Round 1
Reviewer 1 Report
I was invited to revise the paper entitled "Differences in demographics of vaccinees, access to, and satisfaction with SARS-CoV-2 vaccination procedures between German general practices and mass vaccination centers". It was a cross-sectional study aimed to evaluate the difference in level of satisfaction in covid-19 vaccination procedure between patients vaccinated by GPs or vaccination centers, and their related factors, in a sample of German patients.
The topic is relevant and poor studies were performed on this topic. I want to congratulate with Authors for the excellent work.
The study was well structured and clearly reported. Methodology was strong and conclusions are supported by results.
I have only some minor observations:
- Relihability of the questionnaire and the internal consistency should be tested and reported;
- sample size estimation was lacking;
- I suggest to avoid the term "Prediciton" for linear regression analysis. LR is not a prediction model. I suggest to replace it with "Association";
- Lines 278-281: I also suggest to highlight the significant difference in education level. Tertiary educated patients more frequently live in urban area and preferred to take jab in vax centers.
Reviewer 2 Report
This study examined an important clinical issue during the COVID-19 pandemic. The results provided knowledge to develop vaccine uptake programs. The study design and statistical methods were reasonable.
I have some suggestions for the authors.
First, in some countries there were differences in the kinds of vaccines provided and eligibility of vaccinees between general practices and mass vaccination centers. The authors need to explain whether there were differences existing in German.
Second, the authors used several methods to test the adequacy of regression models. Did the authors report the results?
Third, the authors applied statistical methods to examine the moderating effect of age on the association between pre-existing illness and recommendation of vaccination site (Figure 3). It was a good try. More explanation for the statistical method and result can increase readers’ understanding. It is with the results of Figure 4 as well.
